# Microcalcifications Drive Breast Cancer Occurrence and Development by Macrophage-Mediated Epithelial to Mesenchymal Transition

**DOI:** 10.3390/ijms20225633

**Published:** 2019-11-11

**Authors:** Manuel Scimeca, Rita Bonfiglio, Erika Menichini, Loredana Albonici, Nicoletta Urbano, Maria Teresa De Caro, Alessandro Mauriello, Orazio Schillaci, Alessandra Gambacurta, Elena Bonanno

**Affiliations:** 1Department of Biomedicine and Prevention, University of Rome “Tor Vergata”, Via Montpellier 1, 00133 Rome, Italy; manuel.scimeca@uniroma2.it (M.S.); marydecaro12@gmail.com (M.T.D.C.); orazio.schillaci@uniroma2.it (O.S.); 2San Raffaele University, Via di Val Cannuta 247, 00166 Rome, Italy; 3Fondazione Umberto Veronesi (FUV), Piazza Velasca 5, 20122 Milan, Italy; 4Saint Camillus International University of Health Sciences, Via di Sant’Alessandro, 8, 00131 Rome, Italy; 5Department of Experimental Medicine, University “Tor Vergata”, Via Montpellier 1, 00133 Rome, Italy; bonfiglio.rita@gmail.com (R.B.); erika.menichini@gmail.com (E.M.); alessandro.mauriello@uniroma2.it (A.M.); gambacur@uniroma2.it (A.G.); 6Department of Clinical Sciences and Translational Medicine, University of Rome “Tor Vergata”, 00133 Rome, Italy; albonici@med.uniroma2.it; 7Nuclear Medicine, Policlinico “Tor Vergata”, 00133 Rome, Italy; n.urbano@virgilio.it; 8Istituto di Ricovero e Cura a Carattere Scientifico (IRCCS) Neuromed, 86077 Pozzilli, Italy; 9“Diagnostica Medica” and “Villa dei Platani”, 83100 Avellino, Italy

**Keywords:** microcalcifications, breast cancer, osteoblast, BOLCs, EMT

## Abstract

Background: This study aims to investigate: (a) the putative association between the presence of microcalcifications and the expression of both epithelial-to-mesenchymal transition and bone biomarkers, (b) the role of microcalcifications in the breast osteoblast-like cells (BOLCs) formation, and (c) the association between microcalcification composition and breast cancer progression. Methods: We collected 174 biopsies on which we performed immunohistochemical and ultrastructural analysis. In vitro experiments were performed to demonstrate the relationship among microcalcification, BOLCs development, and breast cancer occurrence. Ex vivo investigations demonstrated the significant increase of breast osteoblast-like cells in breast lesions with microcalcifications with respect to those without microcalcifications. Results: In vitro data displayed that in the presence of calcium oxalate and activated monocytes, breast cancer cells undergo epithelial to mesenchymal transition. Also, in this condition, cells acquired an osteoblast phenotype, thus producing hydroxyapatite. To further confirm in vitro data, we studied 15 benign lesions with microcalcification from patients that developed a malignant condition in the same breast quadrant. Immunohistochemical analysis showed macrophages’ polarization in benign lesions with calcium oxalate. Conclusions: Altogether, our data shed new light about the role of microcalcifications in breast cancer occurrence and progression.

## 1. Introduction

Several major pathological conditions, such as cardiovascular diseases and cancer display minerals or organic compound depositions [1]. Tissue calcifications are defined as the deposition of calcium salts, together with smaller amounts of iron, magnesium, and other mineral salts [1].

In breast tissues, microcalcifications play a crucial role in early cancer diagnosis [2]. Indeed, approximately 50% of non-palpable breast cancers are detected by mammography exclusively through microcalcification patterns [3], revealing up to 90% of ductal carcinoma in situ [4]. They can be classified according to their appearance on a mammogram based on the Breast Imaging Reporting and Data system [5], or by their physical and chemical properties [6]. In a recent paper, we demonstrated, for the first time, the presence of magnesium-substituted hydroxyapatite (Mg-HAp), which was frequently noted in breast cancer but never found in benign lesions [7]. In our experience, calcium oxalate (CO) calcification is often associated with benign lesions, whereas hydroxyapatite (HA) is related both to benign and malignant lesions [7]. Recent evidence suggests that the morphological appearance of mammographic microcalcifications is associated with patient prognosis. In fact, patients harboring small breast tumors with casting type calcifications in the mammograms have a poor survival rate for this tumor size category [8,9].

Despite the extensive investigation of breast microcalcifications, the mechanisms leading to their formation are still not defined. Recently, we suggested that ectopic mineralization in pathological conditions might be triggered by the epithelial to mesenchymal transition (EMT) phenomenon and regulated by mechanisms similar to those occurring under physiological conditions [8,10]. Ours and other studies [8,10,11,12,13] allows postulating an overlap between breast microcalcification production and bone mineralization. In particular, Maria Morgan’s group [10,11,12,13] investigated the molecular mechanisms of the microcalcification process in breast cell cultures demonstrating that the mineralization process, related to alkaline phosphatase activity, could be similar to that observed in osteoblast cells. Altogether, these pieces of evidence suggest an active role of microcalcifications in breast cancer occurrence and progression. Thus, this study aims to investigate: (a) the putative association between the presence of microcalcifications in breast lesions and the expression of both EMT and bone biomarkers, (b) the possible role of microcalcifications made of CO in the Breast Osteoblast-Like Cells (BOLCs) formation by macrophage-related EMT, and (c) the putative association between elemental composition of microcalcifications and breast cancer progression.

## 2. Results

### 2.1. Morphological Classification of Breast Lesions

In agreement with the Nottingham classification [14], breast samples were classified as follows: 32 benign lesions (19 fibrocystic mastopathies and 13 fibroadenomas) with microcalcifications (BL+), 20 benign lesions (11 fibrocystic mastopathies and nine fibroadenomas) without microcalcifications (BL−) 74 breast malignant lesions (44 in situ ductal carcinoma and 30 infiltrating ductal carcinomas) with microcalcifications (ML+), and 50 breast malignant lesions (35 in situ ductal carcinoma and 15 infiltrating ductal carcinomas) without microcalcifications (ML−).

### 2.2. Assessment of the Mesenchymal Phenotype in Breast Lesions

In order to assess the acquirement of a mesenchymal phenotype, both in mammary cells and in breast lesions, the expression of vimentin and Cluster of Differentiation (CD44) by immunohistochemistry was assayed. For each sample, immunohistochemical reactions were evaluated by assigning a score from 0 to 3 according to the number of positive breast cells.

Overall, our results showed a significant group effect on the vimentin signal (*p* < 0.0001) (Figure 1A). Also, the expression of vimentin in malignant lesions is significantly higher than in benign lesions (ML− 1.40 ± 0.12 vs. BL− 0.33 ± 0.11; *p* < 0.0001) (Figure 1A–E). Nevertheless, we found that the number of vimentin-positive cells was significantly increased in the lesions with microcalcifications (BL+ 1.00 ± 0.06; ML+ 2.00 ± 0.12) (Figure 1C,E) with respect to those without microcalcifications, both in the benign and in the malignant condition (BL+ vs. BL− *p* < 0.0001; ML+ vs. ML− *p* < 0.0001) (Figure 1B,D).

In line with vimentin expression, we noted a significant group effect about the number of CD44 positive cells (*p* < 0.0001) (Figure 1F–J). In addition, the number of CD44 positive breast cells was significantly higher in malignant lesions when compared to the benignant ones (BL− 0.61 ± 0.11 vs. ML− 1.36 ± 0.07; *p* < 0.0001) (Figure 1G,I). Interestingly, we detected a higher expression of CD44 in the lesions with microcalcifications (BL+ 1.09 ± 0.05; ML+ 1.80 ± 0.11) (Figure 1H,J) when compared to those without microcalcifications, both within the benign and malignant group (BL− vs. BL+, *p* < 0.0001; ML− vs. ML+, *p* = 0.0046).

### 2.3. Identification of Osteoblast-Like Cells in Breast Lesions: BOLCs

Immunohistochemical analysis of Receptor Activator of Nuclear Factor κ B (RANKL), osteopontin (OPN), and Vitamin D Receptor (VDR) was performed to detect the presence of mammary cells with an osteoblast-like phenotype in breast lesions. For each sample, immunohistochemical reactions were evaluated by assigning a score from 0 to 3 according to the number of positive breast cells.

By comparing lesions with microcalcifications and lesions without microcalcifications, we found that RANKL and OPN were consistently more expressed in the lesions with microcalcifications in a significant manner. In more detail, a significant group effect was detected in the rate of OPN positive breast cells (*p* < 0.0001) (Figure 2A–E). OPN expression was focally distributed with an increase in the in the proximity of macrocalcifications (Figure 2C,E). The Mann–Whitney test showed increased OPN expression in BL as compared to both MLs (BL− 0.65 ± 0.07 vs. ML− 1.27 ± 0.10; *p* = 0.0025) (Figure 2A). Moreover, significant difference was observed for OPN expression comparing lesions with (BL+ 1.36 ± 0.17; ML+ 1.70 ± 0.12) or without calcifications (BL− vs. BL+, *p* = 0.0010; ML− vs. ML+, *p* = 0.043). Similarly, significant group effect was observed for RANK-L expression (*p* < 0.0001) (Figure 2F–J). Also, we noted a significant difference between (a) BL− (0.63 ± 0.11) and ML− (1.25 ± 0.06) (*p* < 0.0001) (Figure 2F), (b) BL− and BL+ (1.03 ± 0.29) (*p* = 0.02 )(Figure 2F), (c) ML− and ML+ (1.80 ± 0.06) (*p* < 0.0001) (Figure 2F), and (d) BL− and ML+ (*p* < 0.0001) (Figure 2F).

Despite a significant group effect was observed for the VDR expression (*p* = 0.036) (Figure 3A–E), we did not find differences in the VDR signal, in the presence or absence of microcalcifications (BL+ 1.80 ± 0.10 vs. BL− 1.49 ± 0.13, *p* = 0.084; ML− 1.77 ± 0.12 vs. ML+ 1.92 ± 0.12, *p* = 0.055) (Figure 3A).

### 2.4. Osteoblastic Differentiation and Mineralization in Breast Lesions: A Home for BOLCs

To provide further support to the evidence of osteoblast-like cells in mammary tissue, we wanted to investigate the expression of factors that properly promote the differentiation of osteoblast in bone. Thus, we performed immunohistochemical analysis on Bone Morphogenetics Protein-2 (BMP-2), Bone Morphogenetics Protein-4 (BMP-4), and Pentraxin 3 (PTX3) and we evaluated them by assigning a score from 0 to 3 according to the number of positive breast cells in randomly-selected regions.

Our results showed a significant group effect for all these biomarkers (BMP-2, *p* < 0.0001; BMP-4, *p* < 0.0001; PTX3, *p* < 0.0001) (Figure 3F, and Figure 4A,F). In more detail, we found a striking increase in BMP-2 signal in both benign and malignant lesions with microcalcifications compared to their counterpart with no microcalcifications (BL− 0.65 ± 0.15 vs. BL+ 1.25 ± 0.09, *p* = 0.0011; ML− 1.04 ± 0.09 vs. ML+ 2.13 ± 0.09, *p* < 0.0001) (Figure 3F–J). Interestingly, the intensity of the signal found in the benignant lesions with microcalcifications (BL+) was comparable to that detected in the malignant condition in absence of microcalcifications (ML−) (Figure 3H,I). Consistently, (but at a lower level), the same trend was revealed for the expression of BMP-4 (BL− 0.63 ± 0.16 vs. BL+ 1.14 ± 0.13, *p* = 0.0231; ML− 1.11 ± 0.09 vs. ML+ 1.51 ± 0.10, *p* = 0.0027) (Figure 4A–E). A significant group effect was also found for PTX3 analysis (*p* < 0.0001) (Figure 4F,G). Moreover, we could confirm a significantly higher expression in malignant lesions with microcalcifications (ML+ 2.20 ± 0.11) as compared to malignant lesions without microcalcification (ML− 1.39 ± 0.13) (*p* < 0.0001), as well as in regard to the presence or absence of microcalcifications within the groups of benign lesions (BL− 0.37 ± 0.12 vs. BL+ 1.22 ± 0.23, *p* < 0.0001).

### 2.5. Ultrastructural Characterization of BOLCs

Transmission electron microscopy investigation was performed on both benign/normal (Figure 5A–C) and malignant breast tissue (Figure 5I). Among malignant lesions, we studied the ultrastructural aspect of cells located near microcalcifications. Our analysis confirmed the presence of cells with morphological characteristics typical of osteoblasts (Figure 5D). Their cytoplasm was rich in vesicles containing electron-dense granules similar to the intracellular vesicles of the osteoblasts (Figure 5E,F). Noteworthy, the elemental analysis of the electron-dense bodies inside these vesicles demonstrated the presence of HA (Figure 5E,F). In addition, breast cancer cells close to calcifications were embedded in a matrix rich in collagen (Figure 5G–I).

### 2.6. In Vitro Model of BOLCs Development

Human ex vivo analysis allowed us to develop an in vitro assay to demonstrate the relationship between microcalcification, BOLCs development, and breast cancer occurrence.

#### 2.6.1. Morphological Aspect (Optical and Transmission Electron Microscopy)

Morphological analysis of MDA-MB-231 alone (MDA-MB-231/CTRL) showed that breast cells with mesenchymal characteristics (morphology and vimentin expression) were less than 5%, and no calcifications were present (Figure 6A). Conversely, our experiments demonstrated that a high percent of MDA-MB-231-MΦ/CO underwent mesenchymal transformation and osteoblast differentiation. In particular, at the end of experiments, we observed more than 50% of vimentin-positive spindle-shaped breast cells (Figure 6B). Mesenchymal transformation and osteoblast differentiation of breast cancer cells were also observed in the co-cultures of MDA-MB-231/CO. Nevertheless, in these conditions, we observed to be less than 5% of vimentin-positive breast cells. Noteworthy, we also identified numerous cells with osteoblast characteristics associated with calcification (Figure 6C,D). More important, in the presence of CO (Figure 6C), breast cancer cells increase their proliferation rate, also producing large calcium nodules (Figure 6D).

#### 2.6.2. Western Blot Analysis

Further evidence of cell differentiation are the results obtained by Western blot analysis on protein changes in co-cultures of MDA-MB-231-MΦ/CO when compared with MDA control cells. The mutated form of p53 was expressed in MDA-MB-231/CTRL and in MDA-MB-231/CO, while it was express to undetectable level when the tumor cells were in contact with MDA-MB-231-MΦ/CO, suggesting functional changes typical of a differentiation process (Figure 6E,F).

Another protein taken into consideration is the metabolic marker Pyruvate Kinase isozyme M2 (PKM2). The decrease of its expression indicates a metabolic change from glycolytic (typical of a tumor cell) to oxidative (typical of a differentiated cell). Densitometric analysis after Western blot experiments reveals its decrease in MDA-MB-231-MΦ, in MDA-MB-231-MΦ/HAP but above all in MDA-MB-231/CO (Figure 6F), while the PKM2 expression observed in MDA-MB-231-MΦ/CO is probably due to a low yield of monocytes (Figure 6F).

#### 2.6.3. Transmission Electron Microscopy and Energy Dispersive X-ray (EDX) Microanalysis

TEM analysis, in MDA-MB-231/CTRL, MDA-MB-231/CO, MDA-MB-231-MΦ, MDA-MB-231-MΦ/HAP, showed heterogeneous cell population, including large spindle cells (Figure 6G) and rounded cells (Figure 6H). Noteworthy, in MDA-MB-231-MΦ/CO group, numerous cells showed typical ultrastructural aspect of real osteoblasts such as diameters >50 μm, large cytoplasms rich in vesicles containing electron-dense granules similar to the intracellular vesicle of the osteoblasts, and huge rough reticula (Figure 6I–K). EDX microanalysis demonstrated that electron-dense granules were made of HA (Figure 6J,K).

#### 2.6.4. Progression of Breast Lesions with Microcalcifications: Morphological Classification

Among the totality of the samples collected, in this retrospective study, we further studied 15 benign lesions with microcalcification (foll_BL+) and 15 benign lesions without microcalcifications (foll_BL−) from patients that developed a malignant condition in the same breast quadrant within five years from the first diagnosis (lesions with follow-up, foll_BL). Samples were classified as follows: 15 foll_BL+ consisting of five fibrocystic mastopathies and 10 fibroadenomas, and 15 foll_BL− consisting of six fibrocystic mastopathies and nine fibroadenomas. Concerning the respective malignant lesions, the lesions were classified as showed in Table 1.

#### 2.6.5. Elemental Analysis of Microcalcifications

The EDX microanalysis performed on breast microcalcifications showed that in benign lesions they were mainly composed by CO (12 CO and 3 HA), whereas in the malignant lesions, calcifications were present in 18 tissues: 15 in malignant lesions of patients with previous foll_BL+ (9 HA, 5 Mg-Hap, and 1 CO) and three in malignant lesions of patients with previous foll_BL− (3HA).

#### 2.6.6. The Link among Microcalcifications, Macrophages, and the EMT Phenomenon

EMT was characterized by the immunohistochemistry of vimentin. Vimentin was evaluated by assigning a score from 0 to 3 according to the number of positive breast cells in a randomly selected area containing microcalcifications.

Notably, we found that foll_BL+ showed a significantly higher number of vimentin-positive cells (1.86 ± 0.21) as compared with foll_BL− (0.13 ± 0.10) (*p* < 0.0001) (Figure 7A). Remarkably, we observed that benign lesions with a higher number of vimentin-positive cells tend to develop into more aggressive malignant lesions.

Macrophages were studied as the number of CD68, CD38, or CD163-positive cells on 9.42 mm^2^ of breast tissues (results are expressed as the number of positive cells/9.42 mm^2^ ± SEM) in randomly selected areas (Figure 7B–D). The number of positive CD68 cells was higher in foll_BL+ (101.23 ± 3.57) compared to foll_BL− (10.18 ± 3.52) (*p* < 0.0001) (Figure 7B). Noteworthy, several CD68 positive macrophages were observed close to CO microcalcifications in foll_BL+ (Figure 7B). More important, foll_BL+ lesions were characterized by the presence of M2 macrophages (CD163-positive cells) rather than M1 ones (CD38 positive cells) (CD38: 34.23 ± 4.68 vs. CD163: 64.38 ± 6.20; *p* < 0.0009) (Figure 7C,E,F). No significant differences were observed in the macrophage’s polarization in foll_BL− (*p* = 0.074) (Figure 7D).

## 3. Discussion

Historically, the identification of microcalcifications during mammographic exams is considered a sign of breast disease, both in benign and breast lesions [15]. This makes the mammographic screening the current gold standard clinical methods for the early detection of breast lesions [16]. However, despite the relevant role of microcalcifications in the management of breast cancer patients, and their potential prognostic value, the cellular and molecular mechanisms involved in their formation are largely unknown. In this context, pioneering in vitro studies about osteomimicry of mammary cells have been performed by Maria Morgan and colleagues [11], who demonstrated that bioengineered 3D scaffolds made of collagen glycosaminoglycan support the growth and mineralization of mammary cell lines due to their capability to simulate the bone microenvironment [11]. In line with these pieces of evidence, very recent studies identified molecules involved in breast cancer osteomimicry. In our laboratory, we highlighted some ex vivo data about the microcalcification formation. In fact, we described, for the first time, breast cancer cells capable of producing microcalcifications in a process similar to bone mineralization [7]. Specifically, we showed that breast cancer cells that acquire both morphological and molecular characteristics of mesenchymal cells by the EMT could then assume an osteoblast-like phenotype under the induction of molecules of the bone morphogenetic proteins family [7]. These cells, called BOLCs, show the capability to produce and secrete breast microcalcifications composed by HA [7,8,10,17]. Starting from these pieces of evidence, this study aims to investigate: (a) the putative association between the presence of microcalcifications in breast lesions and the expression of both EMT and bone biomarkers, (b) the possible role of microcalcifications made of CO in the BOLCs formation by macrophages-related EMT, and (c) the putative association between elemental composition of microcalcifications and breast cancer progression.

For as much as the study of the putative association between the presence of microcalcifications in breast lesions and the expression of both EMT and bone biomarkers is concerned, we performed ex vivo study on 174 breast biopsies. Immunohistochemical analysis of vimentin and CD44 confirmed the evidence that in the presence of microcalcifications, both benign and malignant breast lesions are characterized by numerous breast cells with a mesenchymal phenotype. Thus, the presence of microcalcifications could be considered a negative prognostic factor regardless of the type of breast lesion. In this study, we did not carry out analysis of other EMT in situ biomarkers, such as the loss of e-cadherin or the acquisition of the n-cadherin since the data associated with these molecules in breast cancer are still controversial. Indeed, despite numerous studies reporting on the loss of e-cadherin during EMT [18,19], Hollestelle et al. showed that the loss of e-cadherin is not a necessity for EMT in human breast cancer lines [20]. The loss of e-cadherin is well described for a special type of breast cancer—the lobular carcinomas [19]. In addition, Canas-Marques et al. reported several pitfalls in the interpretation of e-cadherin by immunohistochemistry [21]. As reported above, in our case section, no lobular carcinomas were present. Similarly, despite some studies reported the expression of n-cadherin during breast cancer progression, its expression seems to be associated with special types of breast cancer such as lobular carcinomas and micropapillary carcinomas [22,23]. Lastly, in our experience, the number of n-cadherin-positive ductal breast cancer cells is often negligible if evaluated by immunohistochemistry.

More important, ex vivo immunohistochemical analysis showed the prevalence of BOLCs in the malignant breast lesions with microcalcifications when compared to lesions without calcifications. Specifically, we tested the expression of some known osteoblast biomarkers such as RANKL, OPN and VDR [24,25,26,27,28]. Among these RANKL and OPN-positive breast cells were higher in lesions with microcalcifications rather than lesions without microcalcifications. Specifically, OPN and RANKL-positive cells were often close to microcalcifications allowing to hypothesize a role of these cells in the production of calcium crystals. In agreement with this, it is known that OPN is a protein involved in the early phases of hydroxyapatite formation [25], whereas RANKL is a biomarker of real osteoblasts involved in the regulation of bone metabolism by the RANK/OPG/RANKL system [28]. More importantly, our data showed a similar trend in the expression of osteoblast differentiation markers, BMP-2, BMP-4, and PTX3 [29,30,31,32,33]. Consequently, the expression of BMP-2, BMP-4, and PTX3 could explain both the origin of BOLCs and the formation of microcalcifications. Indeed, molecules of the BMP family, as well as PTX3, are known for their capability to induce both osteoblast differentiation and activity. In particular, both Mantovani’s and our group demonstrated the involvement of PTX3 in the deposition of bone matrix [31,32,33]. Immunohistochemical data were also supported by ultrastructural investigation in which we described the morphological characterization of breast cells next to microcalcifications. In fact, the morphology of these cells was similar to that of real osteoblasts (large cytoplasm, a huge rough reticulum and cytoplasm’s rich in vesicles containing electron-dense granules composed of HA).

These and previous ex vivo data have been used to develop an in vitro model for the study of the role of microcalcifications in breast cancer development. In particular, our in vitro model was based on the evidence that (a) CO microcalcifications are more frequently associated with benign breast lesions rather than malignant ones [7,8], (b) the presence of CO can induce the macrophage-mediated EMT in epithelial systems [34], and (c) BOLCs can originate from breast epithelial cells under EMT stimuli [7,8,10]. Thus, to demonstrate a possible active role of CO microcalcifications in breast cancer occurrence and progression, we developed a co-culture system in which breast cancer cell lines (MDA-MB-231) were incubated with both CO and human monocytes. Remarkably, already after 10 days, electron microscopy and EDX analysis displayed the presence of breast cancer cells with osteoblast-phenotype and, most important, the presence of HA crystals. No HA crystals were instead observed in cultures of MDA-MB-231/CTRL or MDA-MB-231-MΦ/CO. Immunohistochemical and western blot analysis of cell cultures also demonstrated that in the presence of CO and monocytes, breast cancer cells undergo to EMT becoming able to produce HA crystals. In agreement with this, in the presence of macrophages and CO, we found an increase in the vimentin-positive MDA-MB-231 cells, a decrease of mutated p53, and metabolic changes toward oxidative metabolism (lower expression of PKM2). These preliminary in vitro data allowed us to hypothesize that CO microcalcifications can participate to breast cancer occurrence and development through the recruitment and activation of macrophages. Specifically, in our hypothesis, CO microcalcifications present in the benign breast lesions could be related to macrophage recruitment and consequently, to the expression of transforming growth factor(TGF) β. As known, TGFβ is the most important EMT inducer able to activate the molecular events responsible to the transformation of breast epithelial cells into mesenchymal-like cells [35]. In the presence of osteoblast differentiation factors such as BMPs and PTX3, these mesenchymal-like cells can then differentiate into BOLCs. Thus, the presence of CO in benign breast lesions could trigger both breast carcinogenesis and the formation of calcifications made of HA.

To further corroborate our hypothesis, we retrospectively collected breast biopsies from 30 patients that developed a malignant lesion within five years from a previous diagnosis of benign lesions (for each patient we collected biopsies of both benign and malignant lesions). In more detail, there were microcalcifications in only 15 benign lesions out of the 30 samples. Noteworthy, in all patients with benign lesions associated with microcalcifications made of CO, the subsequent malignant lesion developed in the same quadrant. Moreover, these malignant lesions were characterized by the presence of microcalcifications made of HA. Malignant lesions of patients with previously benign lesions composed of CO calcifications were poorly differentiated carcinomas. Noteworthy, lesions with CO calcifications were rich in M2 macrophages (CD163 positive cells). No similar results were observed for patients without calcifications in benign lesions.

Altogether our data shed new light about the role of microcalcifications in breast cancer progression and occurrence. In particular, in this study, we proposed a model for breast cancer carcinogenesis based on the capability of CO calcifications to induce macrophage-EMT. Last but not least, this study further emphasizes the biological similarities between bone and breast metabolism.

## 4. Material and Methods

### 4.1. Breast Sample Collection

From December 2010 to December 2014, we collected 176 breast samples in total: breast lesions and 124 malignant breast lesions. From each biopsy, paraffin serial sections were obtained to perform histological classification and immunohistochemical analysis. In addition, 1 mm^3^ tissue fragments were used to perform ultrastructural (transmission electron microscopy) and microanalytical (EDX-microanalysis) investigations. This study protocol was approved by the “Policlinico Tor Vergata” Independent Ethical Committee (reference number # 129.18, 26 July 2018). Written patient consent was obtained for each patient.

### 4.2. Histology

After fixation in 10% buffered formalin for 24 h, breast tissues were embedded in paraffin. Three-micrometer thick sections were stained with hematoxylin and eosin (H&E), and the diagnostic classification was blindly performed by two pathologists [36].

### 4.3. Immunohistochemistry of the Paraffin Sections

Briefly, antigen retrieval was performed on 3-μm-thick paraffin sections using EDTA citrate pH 7.8 or Citrate pH 6.0 buffers for 30 min at 95 °C. Sections were then incubated for 1 h at room temperature with the following primary antibodies, anti-vimentin, anti-CD4, anti-OPN, anti-RANKL, anti-VDR, anti-PTX3, anti-BMP2, and anti-BMP-4 (for details see Table 2). Washings were performed with PBS/Tween20 pH 7.6. Reactions were revealed by the HRP-DAB Detection Kit (UCS Diagnostic, Rome, Italy).

### 4.4. Transmission Electron Microscopy (TEM) of Breast Tissues

Small pieces of breast tissue (1 mm^3^) from surgical specimens were fixed in 4% paraformaldehyde (PFA) and post-fixed in 2% osmium tetroxide [37]. After washing with 0.1 M phosphate buffer, the sample was dehydrated by a series of incubations in 30%, 50%, and 70%, ethanol. Dehydration was continued by incubation steps in 95% ethanol, absolute ethanol, and propylene oxide; then, samples were embedded in Epon (Agar Scientific, Stansted Essex CM24 8GF UK) [37]. Eighty µm ultra-thin sections were mounted on copper grids and observed with Hitachi 7100FA transmission electron microscope (Hitachi, Schaumburg, IL, USA).

### 4.5. Energy Dispersive X-ray (EDX) Microanalysis

All breast samples underwent EDX microanalysis. Six-micrometer-thick paraffin sections were embedded in Epon resin as previously described [38], followed by the identification of microcalcifications. Briefly, sections were deparaffinized, hydrated, osmium tetroxide-fixed, dehydrated in ethanol and propylene oxide, and infiltrated in Epon. The embedding capsules were positioned over areas containing previously-identified microcalcifications. Unstained ultra-thin sections of approximately 100-nm-thick were mounted on copper grids for microanalysis. EDX spectra of microcalcifications were acquired with a Hitachi 7100FA transmission electron microscope (Hitachi, Schaumburg, IL, USA) and an EDX detector (Thermo Scientific, Waltham, MA, USA) at an acceleration voltage of 75 KeV and magnification of 12,000×. Spectra were semi-quantitatively analyzed by the Noran System Six software (Thermo Scientific, Waltham, MA, USA) using the standardless Cliff–Lorimer k-factor method [38]. EDX microanalysis apparatus was calibrated using an x-ray microanalysis standard (Micro-Analysis Consultants Ltd., Cambridgeshire, UK).

### 4.6. Calcium Oxalate Synthesis

CO was synthesized as described by Grases et al. [39]. Briefly, 7 L of distilled water was placwd in a crystallizer and heated to 70 °C. Solutions of Na_2_C_2_O_4_ (7.5 × 10^−3^ M) and CaCl_2_ (7.5 × 10^−3^ M) were dropped simultaneously at the same speed (250 mL per h). The slurry was filtered, and the crystals were washed with water and ethanol and then dried at 50 °C with a vacuum for 24 h.

### 4.7. Cell Culture

MDA-MB-231 cells originally derived from a pleural effusion were obtained from the American Type Culture Collection (ATCC, Manassas, VA, USA) and maintained by the Cell and Tissue Culture Core, Lombardi Cancer Center (Reservoir Rd. NW, Washington, DC, USA). Cells were routinely cultured in DMEM high glucose (Sigma-Aldrich, St. Louis, MO, USA) supplemented with 10% fetal bovine serum (FBS).

### 4.8. Monocyte Isolation

Human peripheral blood monocytes (PBMCs) for co-culture experiments were purified by density gradient (Lympholyte-H, Cedarlane) medium following manufacture’s instruction.

After purification, PBMCs were grown in a 24-well (BD Falcon) cell culture plates previously collagenated (rat-tail collagen type I 0.1 mg/mL, BD) in Roswell Park Memorial Institute (RPMI) 1640 (Euroclone, Pero, Milan, Italy) with 10%. FBS, 2 mM L-Glutamine, Penicillin (100 units/mL)/Streptomycin (100 mg/mL). Approximately 40,000 PBMCs were cultured in each well in 700 µL of media. The cells were incubated at 37 °C, and the next day, 20 ng/mL of macrophage colony-stimulating factor (MCSF, Sigma-Aldrich M 9170) was added to induce monocyte activation. After 6–7 days, co-culture experiments were set up.

The coating plate protocol required 0.1 mg/mL collagen type I of rat-tail high-concentration solution (BD Pharmingen, San Jose, CA, USA, cat# 354249) added to the 24 multi-well plates and incubated for 1 h at 37 °C. After incubation, the plates were washed three times with PBS and kept at +4 °C until use.

### 4.9. In Vitro Model for the Development of “Osteoblast-Like Cells”

For each experiment, about 40,000 PBMCs and 10,000 MDA-MB-231 were cultured. The experimental scheme was MDA-MB-231 + calcium oxalate (MDA-MB-231/CO), MDA-MB-231 + calcium oxalate + activated monocytes (MDA-MB-231-MΦ/CO), MDA-MB-231 + hydroxyapatite + activated monocytes (MDA-MB-231-MΦ/HAP). As controls, we used MDA-MB-231 + activated monocytes (MDA-MB-231-MΦ) and MDA-MB-231 alone (MDA-MB-231/CTRL). After 10–12 days, the cells were used for all subsequent experiments.

### 4.10. Protein Extraction and Western Blot Analysis

Cells were homogenized directly into following buffer: Tris 50 mM, NaCl 150 mM, EDTA 10 mM, Triton-X 1%, and centrifugated at 10,000 g for 2 min. Protein concentrations were determined by the Bradford assay. Proteins were resolved by 12% SDS-PAGE, electrotransferred on PVDF membranes (Amersham™ Hybond™, GE Healthcare Life Science, Pittsburgh, PA, USA, cat# 28906837) and blocked with 5% (*v*/*v*) milk /0.1% (*v*/*v*) TBS-T. The blots were probed with the following primary antibodies: mAb mouse anti-beta actin 1:10,000 (Sigma-Aldrich, St. Louis, MO, USA, cat# A5541), mAb mouse anti-p53 1:500 (Santa Cruz Biotechnology, Santa Cruz, TX, USA cat# sc-126) polyAb rabbit anti-PKM2 1:1000 (Cell Signaling Technology, Danvers, MA, USA, cat# 3198).

Membranes were then incubated with the appropriate horseradish peroxidase-conjugated donkey anti-mouse secondary antibody (Jackson Immuno Research, West Grove, PA, USA cat# 715-035-151) 1:15,000 for p53 and 1:20,000 for beta-actin respectively, goat anti-rabbit secondary antibody (Jackson Immuno Research, St. Thomas’ Place, Cambridgeshire, UK, cat# 111-036-047) 1:15,000. Immunodetection was performed by the enhanced chemioluminescence system Western lighting Plus ECL (Perkin Elmer, Waltham, MA, USA, cat# NEL105001EA).

### 4.11. Cell Culture Immunohistochemistry

Cells were plated on poly-l-lysine coated slides (Sigma-Aldrich, cat #P4707) in 24-well cell culture plates and fixed in 4% paraformaldehyde. After pre-treatment with EDTA citrate at 95 °C for 20 min and 0.1% Triton X-100 for 15 min, cells were incubated 1 h with the rabbit-monoclonal anti-vimentin antibody (see Table 2). Washings were performed with PBS/Tween20 pH 7.6. Reactions were revealed by HRP-DAB Detection Kit (UCS Diagnostic, Rome, Italy) and Mayer’s Hematoxylin to stain the nucleus.

### 4.12. TEM and EDX Analysis of Cell Cultures

Cells were fixed in 4% paraformaldehyde, post-fixed in 2% osmium tetroxide and embedded in Epon resin for morphological studies. After washing with 0.1 M phosphate buffer, the sample was dehydrated by a series of incubations in 30%, 50%, and 70%, ethanol. Dehydration was continued by incubation steps in 95% ethanol, absolute ethanol, and hydroxypropyl methacrylate, then samples were embedded in Epon (Agar Scientific, Stansted Essex, UK).

### 4.13. Statistical Analysis

Statistical analysis was performed using GraphPad Prism 5 Software (San Diego, CA, USA). Immunohistochemical data were analyzed by the Kruskal–Wallis test (*p* < 0.0001) and by Mann––Whitney test (*p* < 0.0005).

## 5. Conclusions

The identification of the molecular mechanisms into breast carcinogenesis represents one of the most challenging topics of translational research. In this scenario, the study of the formation of microcalcifications could be considered a key element for understanding both the biology and clinical aspects of a relevant number of breast cancers. Indeed, according to our model, the presence of microcalcifications, also in benign lesions, can assume a prognostic and predictive value as a sign of EMT occurrence and BOLCs formation. In this context, it is important to remember that the presence of BOLCs in primary breast cancer has been associated with an increased risk of developing bone metastatic lesions [1,8,10,17,40,41]. Therefore, our study can contribute to the re-evaluation of the role of microcalcifications in the management of breast cancer patients laying the foundation for the development of clinical analysis capable of identifying the risk of breast cancer occurrence and progression based on the in vivo detection of microcalcifications.

## Figures and Tables

**Figure 1 ijms-20-05633-f001:**
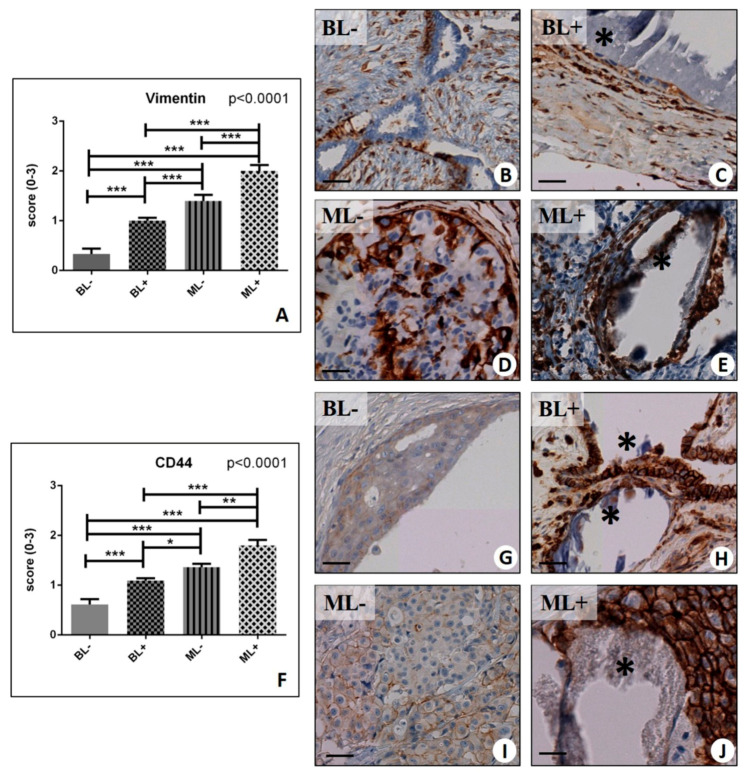
Immunohistochemical evaluation of Vimentin, and Cluster of Differentiation (CD44). (**A**) Graph shows the expression of vimentin in BL−, BL+, ML−, and ML+ groups. (**B**) Image displays vimentin expression in a case of fibroadenoma without macrocalcifications. (**C**) Micrograph showed several vimentin-positive breast cells next to microcalcification (asterisk) in a fibroadenoma. (**D**) Numerous vimentin-positive breast cancer cells in a ductal in situ carcinoma. (**E**) Image shows a calcification (asterisk) surrounded by vimentin-positive infiltrating breast cancer cells. (**F**) Graph shows the expression of CD44 in BL−, BL+, ML−, and ML+ groups. (**G**) No/rare CD44 positive cells in breast fibroadenoma. (**H**) Breast microcalcification (asterisks) in a fibroadenoma surrounded by CD44-positive cells. (**I**) Infiltrating breast carcinomas with rare CD44-positive cells. (**J**) Micrograph displays several CD44-positive breast cancer cells close to the microcalcifications (asterisk) in infiltrating breast cancer. (* *p* < 0.05; ** *p* < 0.01; *** *p* <  0.001). Scale bar represents 100 µm in all images.

**Figure 2 ijms-20-05633-f002:**
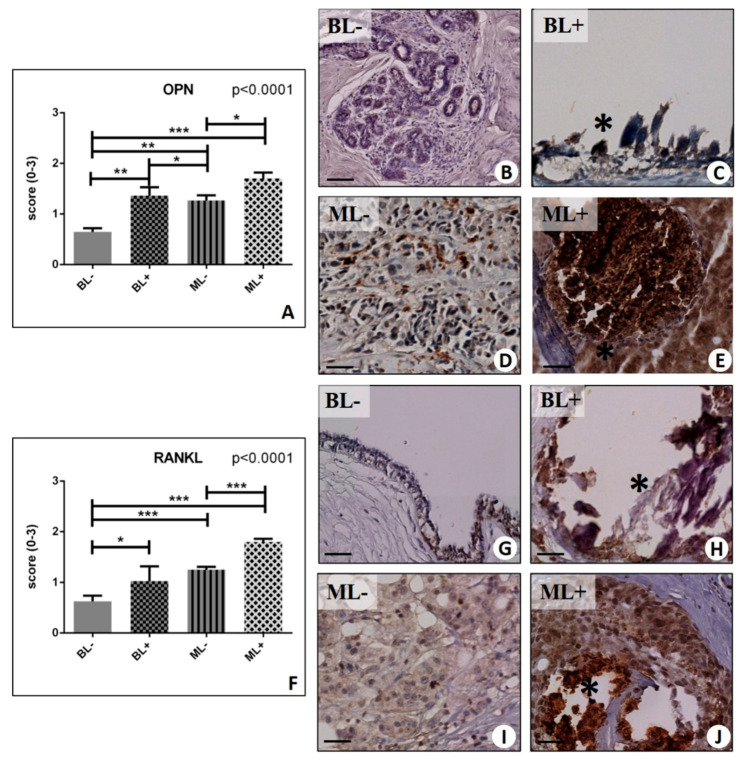
Immunohistochemical evaluation of osteopontin (OPN) and nuclear factor kappa-Β ligand (RANKL). (**A**) Graph shows the expression of OPN in BL−, BL+, ML−, and ML+ groups. (**B**) Breast fibroadenoma with no/rare OPN-positive cells. (**C**) Breast microcalcifications (asterisk) surrounded by OPN-positive cells in a fibroadenoma. (**D**) Several OPN-positive cells in breast infiltrating carcinoma. (**E**) Very high expression of OPN in cells close to microcalcification (asterisk) in an in situ ductal carcinoma. (**F**) Graph shows the expression of RANKL in BL−, BL+, ML−, and ML+ groups. (**G**) Rare RANKL-positive cells in breast fibroadenoma. (**H**) Image shows numerous RANKL-positive breast cell next to calcification (asterisk) in a fibroadenoma. (**I**) Faint staining for RANKL in breast infiltrating carcinoma. (**J**) Several RANKL-positive breast cancer cells next to microcalcifications (asterisk) in infiltrating carcinoma. (* *p* < 0.05; ** *p* < 0.01; *** *p* <  0.001). Scale bar represents 100 µm in all images.

**Figure 3 ijms-20-05633-f003:**
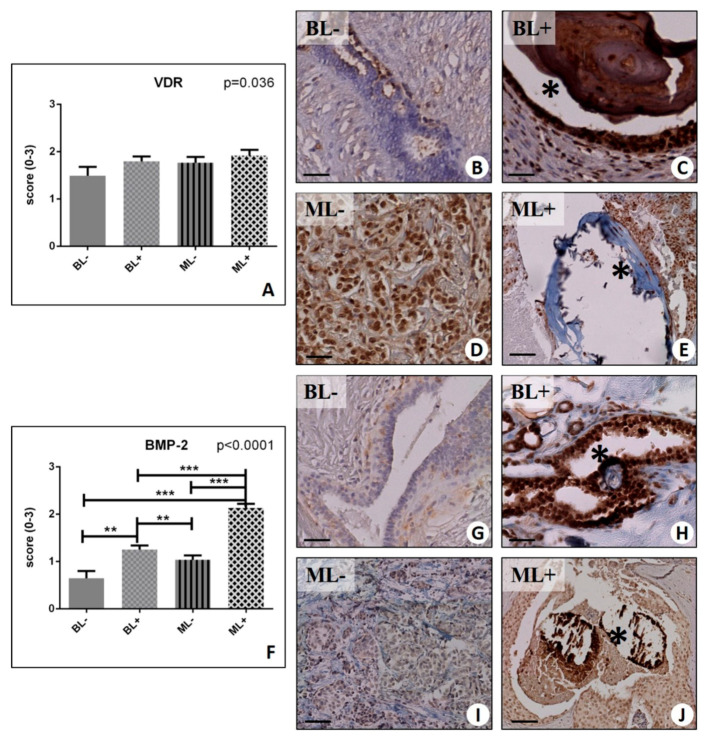
Immunohistochemical evaluation of Vitamin D Receptor (VDR) and Bone Morphogenetic Protein(BMP)-2. (**A**) Graph displays the expression of VDR in BL−, BL+, ML−, and ML+ groups. (**B**) Fibroadenoma with focal nuclear expression of VDR. (**C**) Numerous VDR-positive breast cells in fibroadenoma with a microcalcification (asterisk)(**D**) Several VDR-positive breast cells in infiltrating breast carcinoma. (**E**) Image displays several VDR-positive cells close to microcalcifications (asterisk) in infiltrating carcinoma. (**F**) Graph shows the expression of BMP-2 in BL−, BL+, ML−, and ML+ groups. (**G**) No/rare BMP-2-positive cells in fibroadenoma. (**H**) Numerous BMP-2-positive cells close to microcalcifications (asterisk) in fibroadenoma. (**I**) Image displays faint staining for BMP-2 in breast infiltrating carcinoma. (**J**) Numerous BMP-2-positive breast cancer cells surrounding a calcification (asterisk) in in situ ductal carcinoma. (* *p* < 0.05; ** *p* < 0.01; *** *p* <  0.001). Scale bar represents 100 µm in all images.

**Figure 4 ijms-20-05633-f004:**
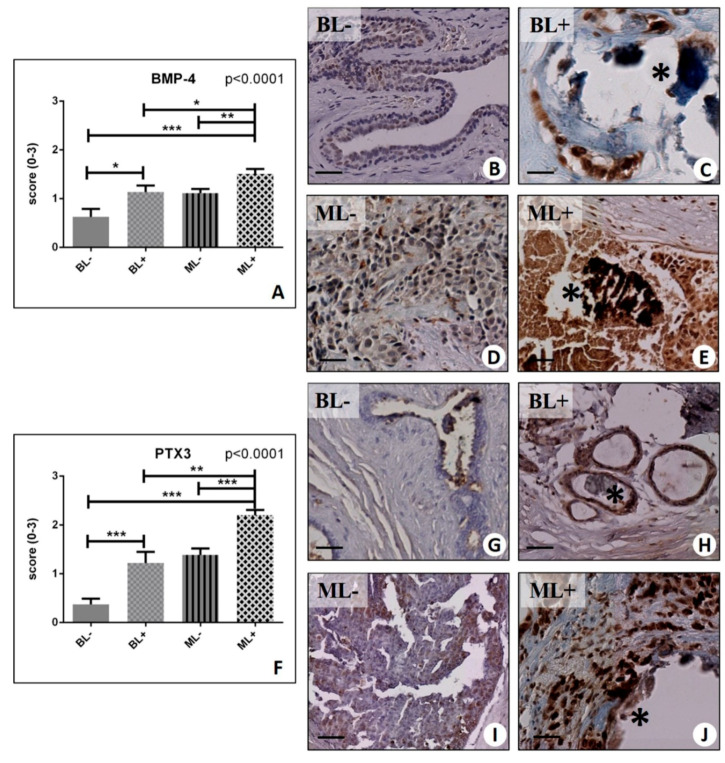
Immunohistochemical evaluation of BMP-4 and Pentraxin 3 PTX3. (**A**) Graph shows the expression of BMP-4 in BL−, BL+, ML−, and ML+ groups. (**B**) No/rare BMP-4-positive cells in fibroadenoma. (**C**) Microcalcification (asterisk) surrounded by BMP-4-positive cells in fibroadenoma. (**D**) Infiltrating breast carcinoma with some BMP-4-positive cells. (**E**) Image shows several BMP-4-positive cells next to calcifications (asterisk) in infiltrating carcinoma. (**F**). Graph shows the expression of PTX3 in BL−, BL+, ML−, and ML+ groups. (**G**) Rare PTX3-positive cells in fibroadenoma. (**H**) Several PTX3-positive cells close to microcalcification (asterisk) in fibroadenoma. (**I**) Infiltrating breast carcinoma with some PTX- positive cancer cells. (**J**) Image displays several PTX3-positive cells close to microcalcification (asterisk) in infiltrating carcinoma... (* *p* < 0.05; ** *p* < 0.01; *** *p* <  0.001). Scale bar represents 100 µm in all images.

**Figure 5 ijms-20-05633-f005:**
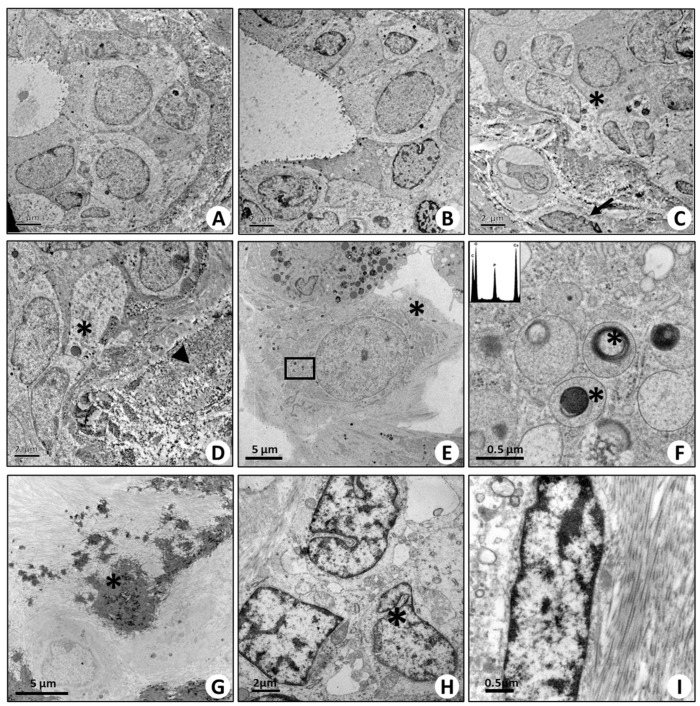
Transmission electron microscopy of breast tissues. (**A**,**B**) Images show the ultrastructure of normal ducts in adjacent areas of fibroadenoma lesions. Both myoepithelial and luminal cells are displayed. (**C**,**D**) Fibroadenoma regions are characterized by epithelial breast cells (arrows), spindle cells (asterisks), and abundant stroma (arrow heads). (**E**) Infiltrating breast cancer is characterized by the presence of microcalcifications, and vimentin-, RANKL-, OPN-, BMP-2-, BMP-4-, and PTX3-positive cancer cells show several osteoblast-like cells. Asterisks mark a cell rich in endoplasmic reticulum, with a large nucleus and some electron-dense granules. Arrows mark a large cell with numerous electron-dense granules. (**F**) High magnification of panel E. The image displays electron-dense granules containing hydroxyapatite (asterisks) (energy-dispersive X-ray (Edx) spectrum). (**G**) Calcified nodule (asterisk) in breast infiltrating carcinoma. (**H**) Several cells surrounding the calcified nodule show large nuclei (asterisks) and collagen fibers both in intracellular and extracellular space. (**I**) High-magnification image of a cell next to the calcified nodule in panel (**G**). The electron micrograph shows the presence of fiber collagen in the cytoplasm of this cell.

**Figure 6 ijms-20-05633-f006:**
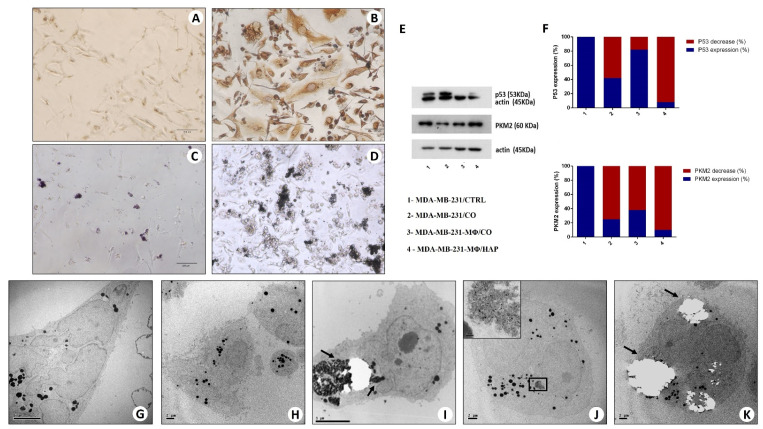
In vitro model of BOLC development. (**A**) Immunocytochemical analysis of MDA-MB-231 alone (MDA-MB-231/CTRL) shows no/few vimentin-positive cells. (**B**) Several vimentin-positive MDA-MB-231 cells are present after co-culture with CO and activated monocytes. (**C**) The image displays the starting point of MDA-MB-231 cells incubated with CO and activated monocytes. (**D**) After 10 days of culture, several large, calcified nodules are present. (**A**–**D**) scale bars represent 100 µm. (**E**) Western blot analysis for tumor protein 53 (p53) and pyruvate kinase muscle (PKM) 2. (**F**) Western blot densitometric analysis (%) of p53 and PKM2 expression compared with the respective controls (MDA). (**G**,**H**) The ultrastructural aspect of MDA-MB-231 (CTRL). (**I**–**K**) Breast osteoblast-like cells with hydroxyapatite intra-cytoplasmatic electron-dense granules (arrows) in MDA-MB-231 + Co +-activated macrophages.

**Figure 7 ijms-20-05633-f007:**
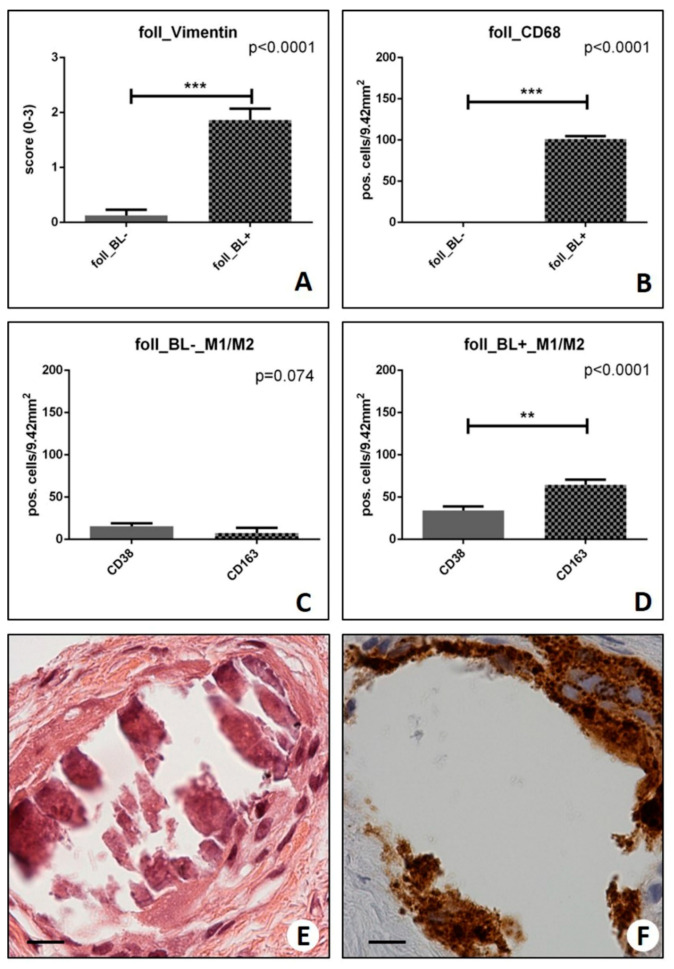
Evaluation of macrophages polarization in breast biopsies. (**A**) Graph shows the expression of vimentin in benign lesions without microcalcification (foll_BL−) and benign lesions with microcalcification (foll_BL+), groups. (**B**) Graph shows the expression of CD68 in foll_BL− and foll_BL+, groups. (**C**) Graph displays the expression of CD68 and CD163 in foll_BL−. (**D**) Graph displays the expression of CD68 and CD163 in foll_BL+. (**E**) H&E staining shows a CO calcification in fibroadenoma. (**F**) Several CD163-positive cells surrounding the breast microcalcification. (** *p* < 0.01; *** *p* <  0.001). Scale bar represents 100 µm in all images.

**Table 1 ijms-20-05633-t001:** List of patients with follow-up.

Patients	Lesion_1	Microcalfification	Lesion_2 (Follow-Up)	Microcalcification
1	fibrocystic mastopathies	Calcium Oxalate	Ductal in situ carcinoma	Hydroxyapatite
2	fibroadenomas	Calcium Oxalate	Infiltrating carcinoma G3	Hydroxyapatite-Mg
3	fibroadenomas	Hydroxyapatite	Ductal in situ carcinoma	Hydroxyapatite
4	fibroadenomas	Calcium Oxalate	Infiltrating carcinoma G2	Hydroxyapatite-Mg
5	fibrocystic mastopathies	Calcium Oxalate	Infiltrating carcinoma G3	Hydroxyapatite-Mg
6	fibrocystic mastopathies	Calcium Oxalate	Infiltrating carcinoma G2	Hydroxyapatite
7	fibroadenomas	Calcium Oxalate	Infiltrating carcinoma G1	Hydroxyapatite
8	fibroadenomas	Hydroxyapatite	Ductal in situ carcinoma	Hydroxyapatite
9	fibroadenomas	Calcium Oxalate	Infiltrating carcinoma G3	Hydroxyapatite-Mg
10	fibrocystic mastopathies	Calcium Oxalate	Infiltrating carcinoma G3	Hydroxyapatite-Mg
11	fibroadenomas	Calcium Oxalate	Ductal in situ carcinoma	Hydroxyapatite
12	fibroadenomas	Calcium Oxalate	Ductal in situ carcinoma	Hydroxyapatite
13	fibroadenomas	Calcium Oxalate	Infiltrating carcinoma G2	Hydroxyapatite
14	fibroadenomas	Calcium Oxalate	Ductal in situ carcinoma	Hydroxyapatite
15	fibrocystic mastopathies	Hydroxyapatite	Ductal in situ carcinoma	Calcium Oxalate
16	fibroadenomas	/	Ductal in situ carcinoma	/
17	fibroadenomas	/	Ductal in situ carcinoma	/
18	fibrocystic mastopathies	/	Ductal in situ carcinoma	/
19	fibrocystic mastopathies	/	Infiltrating carcinoma G1	/
20	fibroadenomas	/	Ductal in situ carcinoma	/
21	fibrocystic mastopathies	/	Infiltrating carcinoma G3	/
22	fibroadenomas	/	Ductal in situ carcinoma	/
23	fibroadenomas	/	Infiltrating carcinoma G1	/
24	fibrocystic mastopathies	/	Infiltrating carcinoma G2	/
25	fibroadenomas	/	Ductal in situ carcinoma	/
26	fibrocystic mastopathies	/	Infiltrating carcinoma G3	/
27	fibroadenomas	/	Ductal in situ carcinoma	/
28	fibroadenomas	/	Infiltrating carcinoma G1	/
29	fibrocystic mastopathies	/	Ductal in situ carcinoma	/
30	fibroadenomas	/	Ductal in situ carcinoma	/

**Table 2 ijms-20-05633-t002:** List of primary antibodies.

Antibody	Characteristics	Reaction Target	Dilution	Retrieval
**anti-Vimentin**	mouse monoclonal clone V9; Ventana, Tucson, AZ, USA	/	Pre-diluted	EDTA citrate pH 7.8
**anti-CD44**	rabbit clone SP37; Ventana, Tucson, AZ, USA	Internal region of human CD44	Pre-diluted	EDTA citrate pH 8.0
**anti-OPN**	Mouse monoclonal clone AE1/AE3/PCK26; Ventana, Tucson, AZ, USA	/	1:100	EDTA citrate pH 7.8
**anti-RANKL**	rabbit monoclonal clone 12A668; AbCam, Cambridge, UK	Membrane bound form of RANKL.	1:100	Citrate pH 6.0
**anti-VDR**	rabbit polyclonal clone NBP1-19478; Novus Biologicals, Littleton, CO, USA	Full-length protein	1:100	Citrate pH 6.0
**anti-PTX3**	rat monoclonal clone MNB1; AbCam, Cambridge, UK	Recognizes the C-terminus of PTX3	1:100	Citrate pH 6.0
**anti-BMP2**	rabbit clone N/A; Novus Biologicals, Littleton, CO, USA	Internal region of human BMP2 (within residues 250-350)	1:500	Citrate pH 6.0
**anti-BMP4**	rabbit polyclonal clone 3C11C7; Novus Biologicals, Littleton, CO, USA	/	1:100	Citrate pH 6.0

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
