# Peer review of "Microcalcifications Drive Breast Cancer Occurrence and Development by Macrophage-Mediated Epithelial to Mesenchymal Transition"

_ijms, 2019, doi:10.3390/ijms20225633_

Round 1

Reviewer 1 Report

The reviewed paper by Scimeca et al. present interesting findings concerning microcalcifications, macrophages and EMI in breast cancer cells. Authors used a range of methods in tumor samples combined with in vitro expriments on MDA-MB-231 cells and human monocytes. There are no serious objections against the apporoach taken, the entire set of clinical samples is also significant.

Some concerns arise, however, with respect to the in vitro experiments and conclusions. For example, it is not clear to the reviewer the criterium of selection of the breast cancer cell line, since the used one is very commonly used for bone metastasis research with corresponding biological characteristics; i.e. invasiveness which would make authors results and conclusions less robust. The question is whether macrophage-driven EMT in such cells does employ the same pathway and mechanism than for example is other breast cancer cells which are not triple negative and invasive per se etc.

Secondly, authors use as a marker of EMT vimentin expression in particular but they should complement it by documented loss of E-cadherin and perhaps N-cadherin profile. EMT is a continuum and many vimentin positive cells are not entirely EMT competent.

Lastly, the language is generally OK but at times it is not easy to read. Also, the use of "group effect" expresison is often inappropriate and should be based on the context rephrased.

Author Response

Reviewer#1

The reviewed paper by Scimeca et al. present interesting findings concerning microcalcifications, macrophages and EMI in breast cancer cells. Authors used a range of methods in tumor samples combined with in vitro expriments on MDA-MB-231 cells and human monocytes. There are no serious objections against the apporoach taken, the entire set of clinical samples is also significant.

Reply: we would like to thank the Reviewer for expressing interest in our work, and for their availability to review our manuscript.

Some concerns arise, however, with respect to the in vitro experiments and conclusions. For example, it is not clear to the reviewer the criterium of selection of the breast cancer cell line, since the used one is very commonly used for bone metastasis research with corresponding biological characteristics; i.e. invasiveness which would make authors results and conclusions less robust. The question is whether macrophage-driven EMT in such cells does employ the same pathway and mechanism than for example is other breast cancer cells which are not triple negative and invasive per se etc.

Reply: Thanks for this point out. The investigation of calcium oxalate in the progression of breast cancers is very innovative. No data is present in literature about the experimental plane developed in this paper. Thus, in this first study, we used a consolidate cancer cell lines for our experiments. We agree with the reviewer about the natural propensity of MDA-MB-231 to develop bone metastasis. However, our experimental plan included several controls that demonstrated that the transformation of breast cancer cells into osteoblast-like cells occurred only in the experimental group MDA-MB-231-MΦ/CO. In addition, we agree with the reviewer about the necessary to confirm these innovative data on several breast cancer cells lines in future studies.

Secondly, authors use as a marker of EMT vimentin expression in particular but they should complement it by documented loss of E-cadherin and perhaps N-cadherin profile. EMT is a continuum and many vimentin positive cells are not entirely EMT competent.

Reply: a very important study of Hollestelle A et al. demonstrated that loss of E-cadherin is not a necessity for epithelial to mesenchymal transition in human breast cancer (Breast Cancer Res Treat. 2013 Feb;138(1):47-57.). Similarly, in breast cancer the expression of N-cadherin is frequently not significative.

Lastly, the language is generally OK but at times it is not easy to read. Also, the use of "group effect" expresison is often inappropriate and should be based on the context rephrased.

Reply:  We performed a complete revision of the manuscript. 

Reviewer 2 Report

A manuscript of highly significant results that shed lights on breast cancer complex studies. A well-written paper, with a logical premise followed by adequately designed methods answering the aims with a justifiable conclusion. The discussion is relatively long and redundant. The pictures are not of high quality and needs extensive labelling of cellular contents. The graphs texts are too small to read, and there are minor typographical errors. Overall, a manuscript of importance that will add to our knowledge and sheds light on the complexity of breast cancer cellular structures and treatment.

Author Response

Reviewer#2

A manuscript of highly significant results that shed lights on breast cancer complex studies. A well-written paper, with a logical premise followed by adequately designed methods answering the aims with a justifiable conclusion.

Reply: we would like to thank the Reviewer for expressing interest in our work, and for their availability to review our manuscript.

The discussion is relatively long and redundant.

Reply: Thanks for this point out. We revised the discussion according to reviewer ‘suggestions.

The pictures are not of high quality and needs extensive labelling of cellular contents.

Reply: Thanks for this point out. We modify the figures according to reviewer ‘suggestions.

The graphs texts are too small to read, and there are minor typographical errors.

Reply: we enlarged the graph and corrected the typographical errors.

Overall, a manuscript of importance that will add to our knowledge and sheds light on the complexity of breast cancer cellular structures and treatment.

Reply: we would like to thank the Reviewer for their comments on how to improve the manuscript

Round 2

Reviewer 1 Report

Authors have addressed the raised concerns and the manuscript is now acceptable for the publishing.

Author Response

we would like to thank the Reviewer for expressing interest in our work, and for their availability to review our manuscript.